# A Review of Recent Studies on Bisphenol A and Phthalate Exposures and Child Neurodevelopment

**DOI:** 10.3390/ijerph18073585

**Published:** 2021-03-30

**Authors:** Machiko Minatoya, Reiko Kishi

**Affiliations:** Center for Environmental and Health Sciences, Hokkaido University, Kita 12, Nishi 7, Sapporo, Hokkaido 060-0812, Japan; mminatoya@cehs.hokudai.ac.jp

**Keywords:** bisphenol A, phthalate, neurodevelopment, developmental disorder

## Abstract

Purpose of Review: Bisphenol A and phthalate have been found in the environment, as well as in humans. In this narrative review pre- and postnatal bisphenol A and phthalate exposures, their relationship to neurodevelopment, and the behavioral outcomes of children are elucidated, focusing in particular on the recent case-control, cross-sectional, and longitudinal studies. This review also introduces some of the possible mechanisms behind the observed associations between exposures and outcomes. Recent Findings: Although bisphenol A and phthalate exposure have been reported to influence neurobehavioral development in children, there are various kinds of test batteries for child neurodevelopmental assessment at different ages whose findings have been inconsistent among studies. In addition, the timing and number of exposure assessments have varied. Summary: Overall, this review suggests that prenatal exposure to bisphenol A and phthalates may contribute to neurobehavioral outcomes in children. The evidence is still limited; however, Attention Deficit Hyperactivity Disorder (ADHD) symptoms, especially among boys, constantly suggested association with both prenatal and concurrent exposure to bisphenol A. Although there is limited evidence on the adverse effects of prenatal and postnatal bisphenol A and phthalate exposures provided, pregnant women and young children should be protected from exposure based on a precautionary approach.

## 1. Introduction

Endocrine-disrupting chemicals (EDCs) have become a priority research area in association with their impacts on human health. Over the last few decades, public awareness of EDCs such as bisphenol A and phthalate has grown significantly as these environmental chemicals have been ubiquitous and can be readily found in the environment, such as in air, river water, and the general population of different age groups [1,2]. Bisphenol A and phthalates are widely used in food packaging and household items, among other products [3]. Bisphenol A is used in polycarbonate products and epoxy resins as coatings on the inside of many food and beverage cans [4]. High-molecular-weight phthalates, including di-2-ethylhexyl phthalate (DEHP), butylbenzyl phthalate (BBzP), di(n-octyl) phthalate (DOP), diisononyl phthalate (DiNP), and diisodecyl phthalate (DiDP), are used in food containers, flooring, wall coverings, and in medical tubing. Low-molecular-weight (LMW) phthalates, such as dimethyl phthalate (DMP), diethyl phthalate (DEP), and dibutyl phthalate (DBP), are used in personal care products or the coating of some medications [5,6]. Exposure can occur via inhalation, ingestion, or dermal contact.

There is particular concern regarding the potential impact of gestational exposure to bisphenol A and phthalates on child neurobehavioral development [7]. During the gestational period, any exposure to EDCs may cause an alteration to the fetus, as many of these chemicals can cross the placenta [8]. Findings from animal studies suggest that the effect of EDC exposure on developmental and neurobehavioral outcomes may have critical periods [9,10]. As bisphenol A and phthalates are EDCs, they may have an impact on the development of sexually dimorphic brain structures prenatally. In laboratory studies, bisphenol A has been shown to disrupt brain function and structure [11,12]. It has been proposed that neurodevelopmental disorders and problems may be caused by these EDCs in modern society [13]. In fact, neurodevelopmental disorders have become an increasingly prevalent problem for children in developed countries [13,14,15]. However, some studies have shown adverse effects, while others found no association. Thus, the purpose of this study is to review the available literature on bisphenol A and phthalate exposure and their relationship to neurodevelopment and behavioral outcomes, focusing in particular on the recent study findings and provide updated evidence.

### Brief Review of Studies Prior to the Past 5 Years

The number of articles including the terms “bisphenol A” or “phthalate”, “child” and “neurodevelopment” prior to the past 5 years was only 11, and 18, respectively. (January, 2021, PubMed). However, the number rapidly increased to 31 and 47, respectively in 2016–2020. This indicates that child neurodevelopment in association with environmental chemical exposure has been taken notice recently. Thus, reviewing focused on the recent findings may be more helpful to understand what is known about the association between exposures and child neurodevelopmental outcomes and what are the issues and future directions.

In past review, limited evidence suggests an association between prenatal and childhood exposure to bisphenol A and phthalates and adverse neurobehavioral outcomes in children [16]. In this review, prenatal exposures to phthalates are considered to be of greatest concern although the critical windows of vulnerability for different endpoints remain to be identified. In prospective studies, a variety of domains including neonatal neurological status, infants’ scores on Bayley Scales of Infant Development (BSID), executive function, behavior problems, patterns of play, and social impairment have been reported in association with higher prenatal phthalate exposure, along with inverse associations between the phthalate levels of school-age children and their intelligence and prevalence of behaviors related to ADHD found in cross-sectional studies [16]. There have been sizable epidemiological studies published to evaluate phthalate exposure and child development prior to the past 5 years. A review of 24 articles indicated that children’s exposure to phthalate metabolites (Monobenzyl phthalate (MBzP), Mono(2-ethyl-5-carboxypentyl), phthalate (MECPP), Mono(2-ethyl-5-hydroxyhexyl) phthalate (MEHHP), Mono(2-ethyl-5-oxohexyl) phthalate (MEOHP), Mono(2-ethylhexyl) phthalate (MEHP), Mono (3-carboxypropyl) phthalate (MCPP), Mono-methyl phthalate (MMP), Mono-ethyl phthalate (MEP), Mono-n-butyl phthalate (MBP), Mono-isobutyl phthalate (MiBP), ∑DEHP (DEHP metabolites), ∑DBP (DBP metabolites)) may bring about impairments in the neurodevelopmental processes such as conduct problems, composite and vocabulary scores [17]. By contrast, results of bisphenol A indicated inconsistency of reporting externalizing problem, anxiety, hyperactivity, emotional control, behavioral inhibition and null association and, all the studies reviewed in this previous article were US studies [17]. Moreover, 4 out of 5 studies were based on the same cohort that indicated the necessity for a comprehensive review based on studies not just limited to studies in the US but prospective studies of other regions such as Asia, Europe and Latin Americas. Since exposure route and level are varied as they depend on lifestyle and culture, findings from various population settings should be investigated [18,19].

A narrative review of EDCs exposure and child development found poor developmental outcomes such as decreased IQ (Intelligence Quotient), poorer memory, Autism Spectrum Disorder (ASD), Attention Deficit Hyperactivity Disorder (ADHD) and other behavioral problems due to exposure to certain phthalates and bisphenol A [20]. Review articles of various EDCs exposure and various health outcomes are useful by and large, however, considering the widespread use and ubiquity of bisphenol A and phthalates, review articles particularly focused on these chemicals becomes more of the reader interest. This review focused on EDCs in general including bisphenol A and phthalates, Previous studies were mostly consistent in suggesting poorer outcomes including developmental delay, and disorders among children with higher prenatal or concurrent exposure. However, findings of low exposure levels were not consistent and the direction of the effects and type of phthalates and metabolites varied. In addition, the trimester-specific influence of exposures on outcomes has not been well investigated. Test batteries and the age of the children tested also varied even though the same domain was investigated. Considering these possible reasons, no clear association has been concluded from the literature prior to the past 5 years. Fewer previous studies have been published on prenatal exposure to bisphenol A and their adverse outcomes on child neurobehavioral development compared to phthalate. One of the comprehensive review articles concluded that the evidence regarding bisphenol A neurotoxicity at levels experienced in the general population does not provide a consistent pattern in the specific behavioral domains most affected [16].

## 2. Methods

To provide up to date evidence and findings, the literature was limited to that published in the past 5 years and written in English. The search was conducted initially in December 2019 and an additional search in December 2020 using the PubMed, Web of Science, and Google Scholar database. The terms included “BPA” or “bisphenol A” and “neurodevelopment” or “neurodevelopmental disorder”. Similarly, terms included “phthalate” and “neurodevelopment” or “neurodevelopmental disorder.” Of these, 24 articles for bisphenol A and 30 articles for phthalate in association with neurodevelopmental outcomes were included.

### 2.1. Bisphenol A and Neurodevelopment

Seventeen longitudinal studies, 3 cross-sectional, and 4 case-control studies were found among 24 articles in this review (Table 1). 

### 2.2. Behavioral Outcomes (Pre-, Postnatal and Concurrent Exposure)

Studies examining behavioral and social-emotional development found that prenatal bisphenol A exposure may be associated with externalizing and internalizing problems [15], anxiety and depression [16], and somatizing behaviors [17] among boys. Contrary, there reported that externalizing problems were significantly associated with prenatal bisphenol A exposure among girls [27]. In addition, prenatal bisphenol A exposure was associated with social impairment [24] and the social domain of Developmental Quotient (DQ) only among girls. Not only prenatal but also postnatal l bisphenol A exposure was associated with increasing difficulties in inhibitory self-control among girls [36]. These studies suggested that pre- and postnatal bisphenol A exposure may affect child behavioral outcomes in a sex-specific manner.

Studies on concurrent exposure to bisphenol A and behavioral outcomes reported both adverse and null effects in children of school age [38,39]. In addition to the nature of the cross-sectional study design that precluded the causal relationship between exposure and behavioral outcomes, an insufficient number of observational studies have been conducted to date to clarify the association of concurrent exposure with child behavioral development.

### 2.3. Cognitive Development (Prenatal Exposure)

Findings from studies examining cognitive and communication development, including intelligence and language development, have been also inconsistent. Studies concluded that prenatal bisphenol A exposure did not affect cognitive development up to 4 years of age [21] and visual-spatial abilities at 8 years of age [30]. Prenatal bisphenol A exposure was associated with reduced full-scale IQ, verbal comprehension [28], and vocabulary scores [26] only among boys, contrary, adverse effects on the language domain of DQ [32] and full-scale IQ, perceptual reasoning, and working memory [26] were reported only among girls. Recent studies suggested association between prenatal exposure and reduced Full Intelligence Quotient (FIQ) [35], and semester specific exposure effect on reduced mental development index (MDI) score at early age [34]. Overall, possible adverse effects of prenatal bisphenol A exposure on cognitive and communication development have been suggested; however, more studies are needed to elucidate this association since these studies only provided limited association in terms of sex and child age.

### 2.4. Psychomotor Development (Prenatal and Concurrent Exposure)

Regarding psychomotor development, reduced psychomotor development at an early age, which did not persist at an older age [21], and no adverse effects at an early age [25,28] have been reported. The influence of prenatal exposure on child psychomotor development was null or weak according to previous studies.

ADHD symptoms, especially among boys, were suggested to be associated with both prenatal and concurrent exposure to bisphenol A in case-control, cross-sectional, and longitudinal studies [20,40,41]. However, recent case-control study found no significant difference in urinary bisphenol A levels of children between ADHD cases and controls [44]. Case-control studies found that diagnosed ASD was associated with higher urinary bisphenol A levels compared with controls [43]. ASD is a multi-factorial disease, however, confounding factors such as maternal and paternal age or education were not adjusted. There might be selection bias in the controls due to difficulty in the recruitment of study participants.

### 2.5. Exposure Assessment of Bisphenol A

Maternal urine samples were used for exposure assessment in most of the studies, except a few. Exposure levels among studies were not simply comparable due to different methods of adjustment, and values were provided in geometric mean, median, and mean. In this review, creatine adjusted concentrations were within the range of 1.05 to 2.6 μg/g Cr and 0.8 ng/mL to 1.9 μg/L for non-creatinine adjusted. By contrast, concentrations in child urine samples showed a wider range from 1.6 ng/mL to 5.28 μg/L. Although most of the studies considered sociodemographic covariates, associations with dietary sources of bisphenol A exposure were not considered, this could be a reason for the wide range of child urine bisphenol A concentration. Additional studies are required to identify potentially modifiable sources of bisphenol A exposure. Studies used cord blood samples for exposure assessment presented varied concentrations from 0.051 to 3.3 ng/mL. Regarding bisphenol A measurement using blood samples, it can possibly be overestimated due to external contamination.

In summary, prenatal and concurrent exposures to bisphenol A were suggested to be associated with child neurobehavioral development of various kinds. This review indicated relatively firm evidence of prenatal bisphenol A exposure and child behavioral emotional problems. In contrast, it demonstrated no or weak effects on psychomotor development. All other developmental outcomes discussed in this review have yet to reach conclusive findings. It also should be noted that studies used different type of specimen collected at different period, which made it difficult to compare exposure levels among studies and to discuss the threshold level of affecting child developmental outcomes.

### 2.6. Phthalates and Neurodevelopment

Nineteen longitudinal studies, 3 cross-sectional studies, and 7 case-control studies were found among 29 articles in this review (Table 2). 

A fair number of longitudinal studies were found, and a wide variety of neurobehavioral outcomes were assessed at various age ranges. The Bayley Scales of Infant Development (BSID) is widely used in longitudinal studies to assess cognitive and psychomotor development in young children [46,47,50,52,56]. Among the 6 studies that used BSID, 2 studies did not find any significant adverse effects of prenatal exposure to phthalates and cognitive, mental, and psychomotor development of children [46,47]. Three studies found adverse effects of prenatal phthalate exposure and outcomes. A study of urban children in the US reported a sex-specific association between maternal urinary MCPP and metabolites of DBP and lower mental development index (MDI) among girls. However, the opposite association was observed among boys [50]. Similarly, prenatal exposure to DBP and BBzP was significantly associated with language delay in children in the US and Sweden [53]. The adverse mental and psychomotor development of children in association with maternal exposure to DEHP was reported in a Korean study [52]. A study using different test batteries also found that the verbal comprehension index of children was negatively associated with urinary MEOHP and the sum of DBP metabolites [49]. A recent study found that the concentration of mono-n-butyl phthalate (MnBP) and sum of DBP metabolites during pregnancy was associated with decreased Psychomotor Developmental Index (PDI) scores in all children. A negative trend of association between exposure to high-molecular-weight (HMW) phthalates and PDI scores was observed in girls, while a positive association was found in boys [56]. Contrary, prenatal exposure to HMW phthalates were inversely associated with motor and cognitive abilities [61]. A longitudinal birth cohort study among Mexican-American children observed negative effects among boys and a positive effect among girls on full-scale IQ in association with sums of maternal HMW and DEHP metabolite levels [54]. Evidence for cognitive intelligence and language development in association with prenatal exposure to phthalates showed discrepancies such as sex effects, however, prenatal DBP exposure constantly has shown negative association with language, verbal, mental and psychomotor development.

Behavioral development and problems were assessed in several studies. Two of the studies that used the Child Behavior Checklist (CBCL) presented consistent findings; maternal DEHP exposure was associated with externalizing problems [45] and significant adverse effects except for somatic complaints in association with maternal DEHP exposure were observed [57]. In addition, other study assessed prenatal exposure to various phthalates and CBCL scores found associations between MBP exposure and anxious-shy behavior among boys and between DBP exposure and psychosomatic problems in both boys and girls [59]. It was reported that prenatal exposure to MCPP also contributed to an increase of internalizing and externalizing problems among children [60]. Various findings have been reported from studies using the Strengths and Difficulties Questionnaire (SDQ). One study observed no significant association between the sum of DEHP metabolite concentrations and child behavior [46]. Another study found that DMP and DnBP were significantly associated with higher total difficulty scores, emotional symptoms, and hyperactivity/inattention problems for DnBP, and peer relationship problems for DMP [62]. The same group further reported that MEP concentration was significantly associated with an increased risk of peer relationship problems [55]. Another study suggested a possible association between MECPP levels in maternal blood and an increased risk of conducting problems [31]. Other studies assessed child behavioral outcomes and found inconsistencies. A study investigating both maternal and paternal phthalate exposure found that maternal preconception urinary MiBP and MCPP concentrations were associated with more total behavioral problems and externalizing behaviors in boys but not girls; whereas paternal preconception urinary MnBP, MiBP, and MBzP concentrations were associated with more internalizing behaviors in boys and less in girls [48]. Two of the cross-sectional studies investigated child behavioral problems associated with current phthalate exposure. Higher levels of MnBP, MEOHP, and MEHHP were associated with an increase in thought problems among girls. Among children aged 6–11 years; significant positive associations were observed between MnBP, MECPP, MEOHP, and MEHHP levels and social problems; and between MnBP, MEOHP, and MEHHP levels and attention problems. However, the same was not observed for older ages [63]. Another study reported that urinary MBzP concentration was significantly associated with emotional symptoms in girls but not in boys aged 6–11 years [39]. Although the types of phthalate metabolites and behavioral problems differed between studies, these 2 cross-sectional studies suggested that some of the behavioral problems in children were associated with current urinary levels of phthalate metabolites. A case-control study suggested that current MEHHP, MEOHP, and MEHP concentrations were related to child attention problems, aggressive behaviors, and externalizing behaviors, while MEHP concentration was related to depressed behaviors and internalizing behaviors [66].

In addition to cognitive, intelligence, and behavioral development, global developmental assessments have been conducted in longitudinal studies. A study assessed the behavior of children at 2 different time points and found that MCPP, MBzP, and the sum of DEHP metabolites were associated with lower levels of smiling and laughing at 12 months. At 24 months, social fear and lower pleasure were linked to higher concentrations of MCPP and MBzP, while higher DEHP was weakly associated with increased anger levels at 24 months [51]. A recent relatively small pilot study found that MMP, MEP, MiBP, and MnBP were significantly associated with increased odds ratios (ORs) of delayed development of all domains (communication, gross motor, fine motor, problem solving, and personal-social) among boys, while most LMW phthalate metabolites and metabolites of DEHP were significantly associated with increased ORs of delayed development of most domains among girls [58]. Finally, Braun et al. reported improved visual-spatial abilities of male children in association with greater prenatal urinary MnBP, MBzP, and MCPP concentrations [30]. In summary, longitudinal studies found adverse, improved, and null effects of prenatal phthalate exposure of various kinds. Results from birth cohort studies have suggested that LMW phthalate such as DBP exposure might increase behavioral problems.

Three case-control studies investigated associations with doctor-diagnosed ADHD and found that ADHD was associated with both prenatal and current phthalate exposure [44,65,66,67]. Higher urinary concentrations of DEHP metabolites, including mono(2-ethyl-5-hydroxyhexyl) phthalate (MEHHP), mono(2-ethyl-5-oxohexyl) phthalate (MEOHP), and mono-(2-ethyl)-hexyl phthalate (MEHP), were dose-dependently associated with ADHD [66]. The study of current exposure and ADHD further examined correlations between phthalate metabolite concentrations, clinical measures, and brain cortical thickness, finding that DEHP metabolites were significantly higher in boys with ADHD than in boys without ADHD [65]. In addition, concentrations of DBP metabolites were significantly higher in the combined or hyperactive-impulsive subtypes than in the inattentive subtype. The metabolite was positively correlated with the severity of externalizing symptoms [65]. Similarly ADHD cases showed higher urinary MnBP levels among boys [44]. The study of prenatal exposure found that the sum of DEHP metabolites was associated with a monotonically increasing risk of ADHD [67]. From the case-control studies, an association between both prenatal and current exposure to DEHP and ADHD was suggested.

Three other 3 case-control studies investigated ASD [43,64,68]. Findings from studies investigating ASD in association with phthalate exposure were inconsistent. A study using phthalate concentrations in house dust found that the levels were not associated with ASD. However, they found that developmental delay (DD) was associated with greater DEHP and BBzP concentrations. Among ASD and DD boys, higher concentrations of DEP and DBP were associated with greater hyperactivity-impulsivity and inattention [64]. Another study conducted by Rahbar et al. focused on a relatively small population; they observed lower MEHHP concentrations and higher MBP concentrations among children with typical development (n = 10) compared to children with ASD (n = 30) [43]. ASD risk was reported to be inversely associated with prenatal exposure to MiBP, mono(3-carboxypropyl) phthalate (MCPP), and mono-carboxyisooctyl phthalate (MCOP) among mothers who took vitamins. However, among mothers who did not take prenatal vitamins, Non-TD risk was positively associated with MCPP, MCOP, and mono-carboxyisononyl phthalate (MCNP) [68]. In their study, MEP, monobenzyl phthalate, MCPP, MCNP, and the sum of DEHP were positively associated with Non-TD risk, but associations with ASD were null among boys, while associations with both ASD and Non-TD were null among girls [68].

In summary, phthalate exposure may be associated with child developmental outcomes and behavioral problems; however, the evidence from previous studies has been inconsistent with respect to different metabolites, the domains that are affected, and sex-specific effects.

### 2.7. Mechanisms Linking Bisphenol A, Phthalate Exposure, and Neurodevelopment

#### 2.7.1. Thyroid Disruption

Thyroid hormones play a critical role in neuronal migration, synaptogenesis, and myelination during gestation and childhood [69]. Phthalate can influence thyroid hormones through biosynthesis, bio-transport, biotransformation, and metabolism [70]. In addition, DEHP can influence thyroid hormones by disturbing the hypothalamus-pituitary-thyroid axis and activating the Ras-Akt-thyrotropin, releasing hormone receptor pathway and inducing hepatic enzymes [71]. DEHP is also suggested to disrupt thyroid function including decreased T4, by damaging thyroid follicles and affecting TTF-1, PAX8, NIS, TPO, and the deiodinase protein family in the recent literature [72].

#### 2.7.2. Brain Function and Structure Disruption

Perinatal exposure to bisphenol A reduces synaptogenesis and synaptic proteins, alters the structure of synapses, affects behavior, and impairs learning-memory in male mice [73,74]. Neonatal and perinatal bisphenol A exposure affects postnatal gene expression and morphology of sexually dimorphic regions in the rat hypothalamus [11,75]. Alternations in the hippocampal dendrites have been reported. In animal studies, gestational exposure to bisphenol A reduces spine density and branching of hippocampal CA1 neurons [76] and decreases spine synapses [77]. Gestational bisphenol A exposure can affect tyrosine hydroxylase immunoreactive neurons, reducing expression in the substantia nigra, locus coeruleus, periventricular preoptic hypothalamus, and midbrain regions [77,78,79,80], although some of these effects are sex-specific.

Phthalate could disrupt dopamine receptor D2, tyrosine hydroxylase, and the homeostasis of calcium-dependent neurotransmitters, reducing dopamine release [81,82,83]. Phthalate metabolites interfere with calcium signaling coupled with nicotinic acetylcholine receptors (nAChRs) in human cell lines, and nAcgRs-mediated calcium channels in the brain and peripheral nervous system play essential roles in a variety of neurodevelopmental processes. In addition, animal studies have suggested that exposure to phthalates results in hippocampal neuron loss and structural and functional alterations [84].

#### 2.7.3. Endocrine and Metabolic Disruption

Bisphenol A was identified as an estrogenic compound acting through estrogen receptors (ERs)-α and β and estrogen-related receptor (ERR)-γ [85,86,87]. It is now known to modulate the androgen receptor (AR) [88]. Bisphenol A can also act via nongenomic intercellular signaling by binding membrane ERs [89] and has been identified as a putative glucocorticoid receptor agonist [90].

From animal and epidemiological studies, phthalates have been suggested to interfere with the homeostasis of sex hormones, including progesterone, androstenedione, and testosterone. Sex hormones may affect child neurodevelopment and behavior via interference of aromatase activity and brain masculinization [84]. Ligands of peroxisome proliferator-activated receptors (PPARs) are known to play roles in lipid metabolism, cellular proliferation, and inflammatory response [91]. Its signal transduction pathway is related to the progression of neurodegenerative and psychiatric diseases, as well as cognitive function [92].

### 2.8. Oxidative Stress

Oxidative stress, a possible chemically induced neuronal damage process, has been studied with respect to these chemical exposures.

Recent laboratory studies have suggested that bisphenol A may increase placental inflammation and disrupt oxidative balance [93]. In epidemiological studies, the effect of bisphenol A on oxidative stress in children with ASD has been investigated, and both studies have shown that bisphenol A may increase oxidative stress, resulting in mitochondrial dysfunction that affects the behavior and functioning of children with ASD [94,95].

Both animal and epidemiological studies have shown that phthalate exposure may affect offspring health by causing oxidative stress. Phthalates have been shown to cause increases in reactive oxygen species and various markers of oxidative stress, potentially via the activation of peroxisome PPARs or by increasing permeability of mitochondrial membranes in a number of in vitro studies [96]. One small cross-sectional study concluded that both bisphenol A and phthalate are associated with increased oxidant stress in healthy children [97]. Regarding the contribution of oxidative stress to bisphenol A exposure and neurobehavioral outcomes, more studies in both laboratories and humans are required.

### 2.9. Epigenetics

According to the Developmental Origins of Health and Disease (DOHaD) theory, epigenetics is suggested to be the underlying mechanism of child neurodevelopment [98]. Prenatal exposure to bisphenol A and phthalates is linked to alterations in DNA methylation and gene expression in the blood of rodent offspring, modulated by an underlying genetic profile [99]. Accumulating evidence suggests that exposure to both bisphenol A and phthalates may contribute to important epigenetic effects [100]. In a recent review article, placental epigenetic consequences such as genomic imprinting, DNA methylation, and the expression of non-coding RNAs in association with maternal exposure to bisphenol A and phthalate were described [101]. A recent study suggested sex-specific effects on placental epigenetic changes in bisphenol A exposure [102]. The role of leptin in behavioral development has also been suggested in previous studies [103,104,105]. A study further investigated leptin produced by the placenta, which is epigenetically regulated by promoter DNA methylation and newborn neurodevelopment. This study suggested that increased placental leptin DNA methylation may play a role in human newborn neurodevelopment, particularly in reactivity to various stimuli in a sex-specific manner [106].

## 3. Future Directions

In this narrative review, bisphenol A and phthalate exposure on child neurodevelopmental outcomes were investigated in the literature published in the past 5 years. Possible reasons for the heterogeneity of findings among studies are exposure misclassification due to relying on a single exposure assessment and/or differences in evaluation of methods (diagnosis vs. parent report, various test batteries, etc.) across the studies. There has been no clear evidence to show which test tool provides more accurate measurements of child neurobehavior. Different adjustment methods for urinary concentration (creatinine vs. specific gravity) may raise the issues of possible misclassification. Reliance on a single spot urine specimen should be considered as well. Reproducibility of bisphenol A or phthalate excretion in single urine specimens is conflicting based on previous literature [107,108]. A study reported poor reproducibility between bisphenol A exposure measured one to three years apart [109]. On the other hand, shorter duration studies either found poor or moderate correlations [110,111,112,113,114]. In addition, diet and time of day the urine/blood specimens collected should be taken into account to investigate neurodevelopmental outcomes. For non-persistent chemicals, including bisphenol A and phthalates, it is difficult to characterize exposure. Lack of understanding of physiological characteristics and pharmacokinetic issues also complicates the interpretation of exposure assessment using bio-samples. Considering the short half-lives of bisphenol A and phthalates, exposure assessment of several time points, as well as standardized assessment procedures such as using the same kind of bio-samples, and the methodology would be desirable in terms of comparing findings across the studies. In addition, factors that may influence the concentrations of biomarkers, such as urinary flow rate, glomerular filtration rate, and pregnancy weight gain, should be taken into account as they may be confounders in epidemiological studies. It should be noted that exposures may be highly correlated within and between different classes of chemicals due to common sources and routes [115]. Due to their high correlation, it is difficult to identify which chemicals or metabolites are responsible for certain effects. For evaluating multiple exposures, applying the exposome concept may be helpful in future studies, although it will be a big challenge, especially for prenatal exposures.

In this review, most longitudinal studies have only investigated prenatal exposures. New evidence has been added from a number of prospective cohort studies of the general population evaluating prenatal exposures and their influence on child neurobehavioral development; repeated follow-ups during childhood should be accelerated in these prospective studies. Exposure to bisphenol A and phthalates are ubiquitous in the postnatal period; moreover, early childhood is a critical period, especially for neurodevelopment [116]. Thus, studies focusing on early to school age exposure to these chemicals are key to understanding child neurodevelopment in association with postnatal exposure.

## 4. Conclusions

In conclusion, for both bisphenol A and phthalates, prenatal exposures were suggested to be associated with child neurobehavioral development. However, evidence is still mostly based on a limited number of studies with large inconsistencies. This review suggested potential harm from bisphenol A and phthalate exposure, as well as the consequences of adverse child neurodevelopment. Although the current review provided only limited evidence to show the adverse effects of prenatal and postnatal bisphenol A and phthalate exposures, pregnant women and young children should be protected from exposure to these chemicals based on a precautionary approach.

## Figures and Tables

**Table 1 ijerph-18-03585-t001:** Bisphenol A exposure and neurodevelopmental outcomes.

Reference	Study Design	Location	Sample Size	Age	Exposure Levels/Specimen	Test Tool	Main Findings
Casas et al. [21]	Longitudinal	Spain	438	1, 4, 7 y	2.6 μg/g Cr (GM)/1st and 3rd trimester maternal urine	BSID-I MSCA DSM-IV CPRS SDQ	Prenatal exposure does not affect cognitive development up to age 4 years. Associations are observed with psychomotor development (β= −4.28, 95% confidence interval (CI): −8.15, −0.41) and ADHD-related symptoms (IRR(Incidence rate ratio) = 1.72; 95% CI: 1.08, 2.73) at early ages, but these do not appear to persist until later ages.
Roen et al. [22]	Longitudinal	US	250	7–9 y	1.9 μg/L (GM)/3rd trimester maternal urine 3.2 μg/L (GM)/child urine	CBCL	High prenatal concentration was associated with increased internalizing (β = 0.41) and externalizing (β = 0.40) composite scores and with their corresponding individual syndrome scales (boys). High postnatal concentration was associated with increased behaviors on both internalizing (β = 0.30) and externalizing (β = 0.33) composite scores and individual sub scores (girls).
Perera et al. [23]	Longitudinal	US	241	10–12 y	1.93 μg/L (GM)/3rd trimester maternal urine 5.28 μg/L (GM)/child urine	RCMAS CDRS	Prenatal exposure was associated with more symptoms of anxiety and depression (β = 2.83) (boys).
Lim et al. [24]	Longitudinal	Korea	304	4 y	2.0 μg/g Cr (mean)/mid-pregnancy urine 4.9 μg/g Cr (mean)/child urine	K-SCQ	Pre and postnatal exposure were associated with increase in social impairment (β = 58.4%; 95% CI: 6.5, 135.8, β = 11.8%; 95% CI: 0.6, 24.3, respectively) (girls).
Braun et al. [25]	Longitudinal	US	346	1–8 y	2.0 μg/g Cr (median)/maternal urine at 16 and 26 weeks	BASC-2 BSID-II WPPSI-III WISC-IV	Prenatal exposure was associated with more externalizing behaviors (β = 5.9; 95% CI: 1.1, 11) (girls).
Lin et al. [26]	Longitudinal	Taiwan	208 (2 y) 148 (7 y)	2 y 7 y	3.2–3.3 ng/mL (median)/cord blood	CDIIT WICS-IV	Prenatal exposure had adverse effects on full-scale IQ and verbal comprehension index (boys). Prenatal exposure had adverse effects on full-scale IQ, perceptual reasoning index, and working memory index (girls).
Stacy et al. [27]	Longitudinal	US	228	8 y	2.1 ng/mL (median)/maternal urine1.6 ng/mL (median)/child urine	BASC-2 WISC-IV BRIEF	Prenatal exposure was associated with more externalizing behaviors (β = 6.2, 95% CI: 0.8, 11.6) (girls). Concurrent exposure was associated with more externalizing behaviors (β = 3.9, 95% CI: 0.6, 7.2) (boys).
Minatoya et al. [28]	Longitudinal	Japan	285	6 m 18 m 3.5 y	0.051 ng/mL (median)/cord blood	BSID-II K-ABC CBCL	No association between prenatal exposure and child mental or psychomotor ability but was positively associated with development problems score (β = 2.60, 95% CI: 0.15, 5.06).
Braun et al. [29]	Longitudinal	Canada	812	3 y	0.8 ng/mL (median)/maternal urine at 12.1 weeks	WPPSI-III BRIEF-P BASC-2 SRS-2	Prenatal exposure was associated with poorer working memory and more internalizing and somatizing behaviors (β = 0.5; 95% CI: −0.1, 1.1, β = 0.6; 95% CI: 0.0, 1.2, respectively) (boys). Prenatal exposure was associated with poorer SRS-2 scores (β = 0.3; 95% CI: 0, 0.7).
Braun et al. [30]	Longitudinal	US	198	8 y	2.0 μg/g Cr (median)/maternal urine at 16 and 26 weeks	VMWM	Prenatal exposure was not associated with VMWM performance.
Minatoya et al. [31]	Longitudinal	Japan	458	5 y	0.062 ng/mL (median)/1st trimester maternal serum	SDQ	No significant association of prenatal exposure.
Pan et al. [32]	Longitudinal	China	368 (12 m) 296 (24 m)	12 m 24 m	1.05 μg/g Cr (median)/maternal urine at delivery	DQ	Prenatal exposure was adversely associated with the adaptive domain DQs (boys and girls) (β = −1.43; 95% CI: −2.30, −0.56), and the social domain DQs (girls) at 12 mo, as well as with the language domain (girls) at 24 mo (β = −1.69; 95% CI: −3.23, −0.15).
Jensen et al. [33]	Longitudinal	Denmark	535 (MB-CDI) 658 (CBCL)	21 m 2–7 y	1.2 ng/mL (median)/3rd trimester maternal urine	MB-CDI CBCL	Prenatal exposure adversely associated with vocabulary score (boys). BPA exposure in the highest tertile had OR = 3.70; 95% CI: 1.34–10.21.
Jiang et al. [34]	Longitudinal	China	456	2 y	1.13 μg/L (median, average)/1st, 2nd, 3rd trimester maternal urine	BSID-I	Increase in BPA concentrations was related to lower MDI scores only in the 2nd trimester (β = −2.87, 95 % CI: −4.98, −0.75).
Guo et al. [35]	Longitudinal	China	326	7 y	2.78 μg/L (GM)/Maternal urine	C-WISC	Prenatal exposure was significantly negatively associated with FIQ (β = −1.18, 95% CI: −2.21, −0.15).
England-Mason et al. [36]	Longitudinal	Canada	312	2 and 4 y	1.64 μg/g Cr (GM)/3rd trimester maternal urine 1.11 μg/g Cr (GM)/postpartum maternal urine	BRIEF-P	Higher concentrations of maternal BPA at 3-month postpartum were associated with increasing difficulties in inhibitory self-control and emergent metacognition from age 2 to 4 (girls).
Freire et al. [37]	Longitudinal	Spain	191	4–5 y	1.30 ng/g (median)/placenta	MSCA	Prenatal exposure was associated with greater ORs of scoring lower in the verbal (OR = 2.78, 95% CI: 1.00–5.81) and gross motor (OR = 1.75, 95% CI: 1.06–9.29).
Perez-Lobato et al. [38]	Cross-sectional	Spain	300	9–11 y	4.76 μg/L (median)/child urine	CBCL/6-18	Concurrent exposure was associated with worse behavioral scores. Children with the highest BPA had more somatic complaints (β = 2.35; 95% CI: 0.25, 4.46) and social (β = 1.71; 95% CI: 0.19, 3.22) and thought problems (β = 2.58; 95% CI: 0.66, 4.51).
Arbuckle et al. [39]	Cross-sectional	Canada	1080	6–11 y	1.31 μg/L (GM)/child urine	SDQ	Concurrent exposure was not associated with SDQ score but was associated with taking psychotropic medications (OR = 1.59; 95% CI: 1.05–2.40).
Tewar et al. [40]	Cross-sectional	US	460	8–15 y	3.9μg/L(median)/child urine	ADHD diagnosis	Concurrent exposure was associated with ADHD and the association was stronger in boys (OR = 10.9; 95% CI: 1.4–86.0).
Li et al. [41]	Case-control	China	215 (case) 253 (control)	6–12 y	Case 4.63 μg/g Cr Control 1.71 μg/g Cr (mean)/child urine	ADHD diagnosis	Concurrent exposure may be related to ADHD. (OR = 4.58; 95% CI: 2.84–7.37 boys, OR = 2.83; 95% CI: 1.17–6.84 girls).
Stein et al. [42]	Case-control	US	46 (ASD) 52 (control)	10 y (mean)	Case 11.18 ng/mL Control 6.58 ng/mL (median)/child urine	ASD diagnosis	Concurrent exposure was associated with ASD. Total BPA was 3 times greater with the ASD group (*p* < 0.001).
Rahbar et al. [43]	Case-control	US	30 (ASD) 10 (control)	2–8 y	Case 1.33 µg/g Cr Control 0.93 µg/g Cr (mean)/child urine	ASD diagnosis	Controls had lower bisphenol A levels.
Tsai et al. [44]	Case-control	Taiwan	130 (ADHD) 68 (control)	6–12 y	Not provided.	ADHD diagnosis	No significant difference of urinary BPA levels between case and control groups.

ASD Autism Spectrum Disorder, ADHD Attention Deficit Hyperactivity Disorder, CBCL Child Behavior CheckList, SDQ Strengths and Difficulties Questionnaire, BSID Bayley Scales of Infant Development, MSCA McCarthy Scales of Children’s Abilities, DSM Diagnostic and Statistical Manual of Mental Disorders, CPRS Conner’s Parent Rating Scales, RCMAS Revised Children’s Manifest Anxiety Scale, CDRS Children’s Depression Rating Scale, K-SCQ Korean version of the Social Communication Questionnaire, BASC Behavior Assessment System for Children, WPPSI Wechsler Preschool and Primary Scales of Intelligence, WISC Wechsler Intelligence Scale for Children, CDIIT Comprehensive Developmental Inventory for Infants and Toddlers, BRIEF Behavior Rating Inventory of Executive Function, K-ABC Kaufman Assessment Battery for Children, BRIEF-P Behavior Rating Inventory of Executive Function–Preschool, SRS Social Responsiveness Scale, VMWM Virtual Morris Water Maze, DQ developmental quotient, and MB-CDI MacArthur-Bates Communicative Development Inventories.

**Table 2 ijerph-18-03585-t002:** Phthalate exposure and neurodevelopmental outcomes.

Reference	Study Design	Location	Sample Size	Age	Exposure Level/Specimen	Test Tool	Main Findings
Lien et al. [45]	Longitudinal	Taiwan	122	8 y	MBP = 66.88 μg/g Cr MEOHP = 13.59μg/g Cr MEHP =16.93 μg/g Cr (GM)/3rd trimester maternal urine	CBCL	Prenatal DEHP and DBP exposures were associated with externalizing domain behavior problems. MBP (β = 4.29; 95% CI: 0.59, 7.99), MEOHP (β = 3.74; 95% CI: 1.33, 6.15), and MEHP (β = 4.28; 95% CI: 0.03, 8.26).
Gascon et al. [46]	Longitudinal	Spain	367	1, 4, 7 y	MBzP = 11.9 μg/g CrΣ4DEHP = 99.8 μg/g Cr (median)/maternal urine at 12 and 32 weeks of gestation	BSID-I MSCA DSM-IV CPRS SDQ	No associations between prenatal exposure and cognitive and psychomotor scores at the age of 1 and 4 years, except for MBzP levels and lower psychomotor scores (β = −1.49; 95% CI: −2.78, −0.21). Prenatal DEHP exposure was associated with increased social competence scores at 4 years (β = 2.00; 95% CI: 0.22, 3.79) and with reduced ADHD symptoms at age 4 and 7 years (IRR = 0.84; 95% CI: 0.72, 0.98, IRR = 0.83; 95% CI: 0.71, 0.95, respectively). Prenatal DEP exposure was associated with a reduced risk of inattention symptoms at 4 years (IRR = 0.88; 95% CI: 0.80, 0.97).
Minatoya et al. [47]	Longitudinal	Japan	328	6 m 18 m	DEHP = 8.81 ng/mL (median)/maternal blood	BSID-II	Prenatal DEHP exposure was not associated with mental and psychomotor development.
Messerlian et al. [48]	Longitudinal	US	166	2–9 y	ΣDEHP = 60.7 ng/mL MiBP = 5.9 ng/mL (GM)/maternal urine before pregnancy	BASC-2	Pre-conceptional exposure to DEHP was associated with a decrease in internalizing behavior scores (β = −2.0; 95% CI: −3.2, −0.7). MiBP was positively associated with externalizing behaviors (β = 1.7; 95% CI: 0.3, 3.2) (boys).
Huang et al. [49]	Longitudinal	Taiwan	204	3–12 y	MEOHP = 32.2 μg/g Cr MnBP = 46.2 μg/g Cr MiBP = 24.3 μg/g Cr (median)/child urine	WISC-IV WPPSI-R	Postnatal exposure to DEHP and DnBP affect intellectual development, particularly language learning or expression ability (β = −11.92; 95% CI: −22.52, −1.33, β = −10.95; 95% CI: −20.74, −1.16, respectively).
Doherty et al. [50]	Longitudinal	US	276	24 m	MnBP = 33.0 μg/L MnBP = 5.6 μg/L MCPP = 2.9 μg/L (GM)/31st weeks maternal urine	BSID-II	Prenatal exposure to DnOP and DBP were associated with lower MDI scores (girls) and improved scores (boys).
Minatoya et al. [31]	Longitudinal	Japan	458	5 y	MECPP = 0.20 ng/mL (median)/maternal serum	SDQ	Possible association between prenatal DEHP exposure and increased risk of conduct problems (OR = 2.78, 95% CI 1.36–5.68).
Singer et al. [51]	Longitudinal	US	204 (12mo) 279 (24 m)	12 m/24 m	3rd trimester Maternal urine	IBQ (12mo) TBAQ (24mo)	Prenatal exposure to DnOP and BBzP were associated with social fear (β = 0.3; 95% CI: −0.1, 0.6, β = 0.3; 95% CI: 0.0, 0.5, respectively) and lower pleasure β = −0.2; 95% CI: −0.4, −0.1, β = −0.1; 95% CI: −0.2, 0.0, respectively) at 24 mo.
Braun et al. [30]	Longitudinal	US	198	8 y	MnBP = 24 μg/g Cr (median)/maternal urine at 16 and 26 weeks	VMWM	Prenatal higher MnBP was associated with longer time (1.7 s; 95% CI: −0.7, 4.1) and shorter distance (−1.7 units; 95% CI: −2.8, −0.5) (girls), and with shorter time (−3.0 s; 95% CI: −5.6, −0.4) (boys).
Kim et al. [52]	Longitudinal	Korea	140	1–2 y	MEP = 11.9 μg/g Cr (median)/maternal urine at delivery MEHP = 2.5 μg/L (median)/breast milk 30days after delivery	BSID-II SMS CBCL	Prenatal DEP exposure was associated with early mental (β = −2.40; 95% CI: −4.39, −0.40), psychomotor (β = −2.25; 95% CI: −4.03, −0.47), and social development (β = −2.54; 95% CI: −4.44, −0.65). Postnatal exposure to DEHP was inversely associated with mental (β = −5.60; 95% CI: −11.05, −0.14).
Bornehag et al. [53]	Longitudinal	US, Sweden	963 (Sweden) 370 (US)	30 m (Sweden) 37 m (US)	MBP = 69.4 (Sweden), 6.5 (US) ng/mL MBzP = 16.1 (Sweden), 3.4 (US) ng/mL (GM)/maternal urine at 10.9 weeks	Language Development Assessment	Prenatal exposure to DBP (OR =1.29; 95% CI: 1.03, 1.63) and BBzP (OR = 1.14; 95% CI: 1.00, 1.31) was associated with language delay.
Hyland et al. [54]	Longitudinal	US	334	7–16 y	ΣLMW = 1.5 nmol/mL (GM)/maternal urine	BRIEF WCST WICS-IV ENI NEPSY-II SRS-2 BASC-2 SRP CADS CPT-II	Prenatal exposure to LMW phthalates was associated with more self-reported hyperactivity (β = 0.8; 95% CI: 0.1, 1.4), attention problems (β = 1.5; 95% CI: 0.7, 2.2), and anxiety (β = 0.9; 95% CI: 0.0, 1.8) at age 16.
Jankowska et al. [55]	Longitudinal	Poland	134	7 y	MEP = 19.4 μg/g Cr MnBP = 4.1 μg/g Cr Oxo-MEHP = 1.6 μg/g Cr (median)/maternal urine at 3rd trimester	SDQ IDS	Prenatal exposure to DEP was associated with an increased risk of peer relationship problems (OR = 2.7). Prenatal DEP and DnBP exposure were negatively associated with fluid intelligence (β = −5.2, β = −4.9, respectively) and cognition (β = −4.2, β = −4.0, respectively) while the oxo-MEHP level was positively associated (β = 3.6, β = 2.9, respectively).
Qian et al. [56]	Longitudinal	China	476	2 y	ΣLMW = 547.96 nmol/L ΣHMW = 107.29 nmol/L (median)/maternal urine	BSID-II	Prenatal exposure to DBP was associated with decreased PDI scores (β = −1.89, 95% CI: −3.63, −0.15). A negative association between exposure to HMW phthalates and PDI scores (girls), as well as a positive association (boys), were found.
Chen et al. [57]	Longitudinal	Taiwan	122/96/78	8,11, 14 y	DEHP = 4.54 μg/kg_bw/day/maternal urine at 3rd trimester	CBCL	Prenatal DEHP exposure was associated with increased CBCL scores (2.02 for internalizing, 2.88 for externalizing problems).
Dong et al. [58]	Longitudinal	China	138	9 m	MMP = 29.98 μg/g Cr MEP = 52.45 μg/g Cr MiBP = 127.49 μg/g Cr MnBP = 115.08 μg/g Cr MEHP = 23.67 μg/g Cr (GM)/maternal urine	ASQ-3	Prenatal exposure was mostly associated with DD. MMP, MEP, MiBP and MnBP levels were associated with increased ORs of DD of all domains (boys), and LMW phthalate and DEHP were associated with increased ORs of DD of most domains (girls).
Daniel et al. [59]	Longitudinal	US	411	7 y	MnBP = 13.3 ng/mL MiBP = 9.1 ng/mL MnBP = 37.4 ng/mL MEOHP = 18.4 ng/mL MEHHP = 22.1 ng/mL (GM)/3rd trimester maternal urine	CBCL, CPRS	Increases in in anxious-shy behaviors were associated with prenatal exposure to MBzP (MR = 1.20, 95% CI 1.05–1.36) and MiBP (MR = 1.22, 95% CI 1.02–1.47) (boys). Increases in perfectionism were associated with MBzP (MR = 1.15, 95% CI 1.01–1.30), decreased hyperactivity was associated with MEOHP (MR = 0.83, 95% CI 0.71–0.98) and MEHHP (MR = 0.85, 95% CI 0.72–0.99) (girls). Increases in psychosomatic problems were associated with MiBP (MR = 1.28, 95% CI 1.02–1.60), and MnBP (MR = 1.28, 95% CI 1.02–1.59) (boys and girls).
Li et al. [60]	Longitudinal	US	314	2, 3, 4, 5, 8 y	Values were provided as log_10_-transformed creatinine-standardized form/maternal urine at 16 and 26 weeks and child urine	BASC-2	Prenatal MCPP exposure was associated with more problem behaviors (internalizing: β = 0.9, 95% CI = −0.1, 1.9; externalizing: β = 1.0, 95% CI = −0.1, 2.0; BSI: β = 1.1, 95% CI = 0.1, 2.1). The weighted childhood phthalate index was associated with more problem behaviors (internalizing: β = 1.5, 95% CI = −0.2, 3.1; externalizing: β = 1.7, 95% CI = 0.1, 3.5; BSI: β = 1.7, 95% CI = 0.2, 3.2); MBzP, MCNP, and MEP largely contributed to these associations.
Torres-Olascoaga et al. [61]	Longitudinal	Mexico	218	48 m	MEHP = 7.8–9.5 ng/L MEHHP = 30.5–32.5 ng/L MEOHP = 16.3–19.4 ng/L MECPP = 46.7–55.4 ng/L MBzP = 4.6–7.3 ng/L MCPP = 1.6–2.4 ng/L (mean)/1st, 2nd, 3rd trimester maternal urine	MSCA	Inverse association was observed between the prenatal exposure to HMW phthalates.
Jankowska et al. [62]	Cross-sectional	Poland	250	7 y	DnBP = 62.6 μg/L MEP = 42.0 μg/L (median)/child urine	SDQ IDS	Prenatal exposure to DMP and DnBP were associated with higher total difficulties scores (β = 1.5; 95% CI: 0.17, 2.7, β = 1.5, 95% CI: 0.25, 2.8, respectively). Prenatal DnBP and DMP exposures were negatively associated with fluid IQ (β = −0.14, 95% CI: −0.29, 0.0041) and crystallized IQ (β = −0.16, 95% CI: −0.29, −0.025). Prenatal DMP (β = −0.17; 95% CI: −0.31, −0.033), DEP (β = −0.16; 95% CI: −0.29, −0.018), and DnBP (β = −0.14; 95% CI: −0.28, 0.0012) exposure were associated with mathematical skills.
Won et al. [63]	Cross-sectional	Korea	1723/867	6–18 y	MBzP = 4.82 μg/g Cr MnBP = 40.84 μg/g Cr MECPP = 32.10 μg/g Cr MEOHP = 15.95 μg/g Cr MEHHP = 21.76 μg/g Cr (GM)/child urine	CBCL ARS	Concurrent exposure to DnBP was associated with social (β=0.60; 95% CI: 0.15, 1.05), thought (β=0.55; 95% CI: 0.08, 1.03), and attention problems (β=0.68; 95% CI: 0.21, 1.14). Higher levels of MnBP, MEOHP, and MEHHP were associated with an increase in thought problems (girls). Among children aged 6–11 years, positive associations between the MnBP, MECPP, MEOHP, and MEHHP levels and social problems, as well as between the MnBP, MEOHP, and MEHHP levels and attention problems, were observed.
Arbuckle et al. [39]	Cross-sectional	Canada	1080	6–11 y	MBzP = 21.23 μg/L (GM)/child urine	SDQ	Concurrent BBzP exposure was associated with emotional symptoms (OR = 1.38; 95% CI: 1.09, 1.75) (girls).
Philippat et al. [64]	Case-control	US	50 (ASD) 27 (DD) 68 (TD)	24–60 m	DEHP = 187 μg/g BBzP = 13 μg/g DEP = 1 μg/g DBP = 10 μg/g (median)/home dust	VABS ABC MSEL	Concurrent DEHP and BBzP levels in indoor dust were higher at homes of DD children (OR = 2.10; 95% CI: 1.10, 4.09, OR = 1.40; 95% CI: 0.97, 2.04, respectively). Among ASD and DD, higher indoor dust levels of DEP and DBP were associated with greater hyperactivity-impulsivity and inattention (boys).
Park et al. [65]	Case-control	Korea	180 (ADHD) 438 (control)	6–15 y	MEHP = 45.60 μg/g Cr MEOP = 43.82 μg/g Cr MBP = 68.03 μg/g Cr (GM)/child urine	ADHD-RS CGI-S DBDS DSM-IV CBCL CPT brain cortical thickness	Concurrent DEHP levels were higher with ADHD than without ADHD (boys). Concurrent DBP levels were higher in the combined or hyperactive-impulsive subtypes compared to the inattentive subtype and was positively correlated with the severity of externalizing symptoms. Concurrent DEHP levels were negatively correlated with cortical thickness.
Hu et al. [66]	Case-control	China	225 (ADHD) 225 (control)	6–13 y	MEHP = 6.72 ng/mL MEHHP = 30.23 ng/mL MEOHP = 14.94 ng/mL (median)/child urine	SNAP-IV CBCL	Concurrent DEHP levels were dose-dependently associated with ADHD (ORs = 2.35–3.04) and co-occurring ODD (ORs = 3.27–4.44) and related to attention problems, aggressivity, depression, and externalizing and internalizing behaviors.
Rahbar et al. [43]	Case-control	US	30 (ASD) 10 (control)	2–8 y	(ASD) MEHHP = 14.94 μg/g Cr (control) MEHHP =11.08 μg/g Cr (mean)/Child urine	ASD diagnosis	Controls had lower MEHHP (26%) and higher MBP (50%) levels compared to the cases.
Engel et al. [67]	Nested case-control	Norway	297 (ADHD) 553 (control)	born 2003–2008	(ADHD) ΣDEHP = 0.31 μmol/L (control) ΣDEHP = 0.27 μmol/L (GM)/child urine	ADHD diagnosis	Prenatal exposure to DEHP was associated with increased the risk of ADHD (OR = 2.99; 95% CI: 1.47, 5.49).
Shin et al. [68]	Case-control	US	46 (ASD) 55 (Non-TD) 100 (TD)	3 y	(ASD) ΣDEHP = 0.14 μmol/L (Non-TD) ΣDEHP = 0.19 μmol/L (TD) ΣDEHP = 0.17 μmol/L (median)/maternal urine during pregnancy	clinical diagnosis	Prenatal exposure to DEP was associated with an increased risk of Non-TD (RRR = 1.38; 95% CI: 1.01, 1.90). ASD risk was inversely associated with prenatal exposure to DiBP, DMP, and DiNP; however, among mothers who did not take prenatal vitamins, (MiBP, RRR = 0.44; 95% CI: 0.21, 0.88), (MCPP, RRR = 0.41; 95% CI: 0.20, 0.83) (MCOP, RRR = 0.49; 95% CI: 0.27, 0.88), the Non-TD risk was positively associated with DnOP, MCOP and DDP exposure (MCPP, RRR = 5.09; 95% CI: 2.05, 12.6), (MCOP, RRR = 1.86; 95% CI: 1.01, 3.39), (MCNP, RRR = 3.67; 95% CI: 1.80, 7.48). Prenatal exposure to DEP, DBP, DiNP, DMP, and DEHP were positively associated with Non-TD risk (boys).
Tsai et al. [44]	Case-control	Taiwan	130 (ADHD) 68 (control)	6–12 y	Not provided.	ADHD diagnosis	ADHD group demonstrated higher MnBP (*p* = 0.014) (boys).

ASD Autism Spectrum Disorder, DD Developmental Delay, TD Typically Developing, VABS Vineland Adaptive Behavior Scales, ABC Aberrant Behavior Checklist, MSEL Mullen Scales of Early Learning, ADHD-RS Attention Deficit Hyperactivity Disorder Rating Scale, CGI-S Clinical Global Severity Scale, DBDS Disruptive Behavioral Disorder Rating Scale, DSM Diagnostic and Statistical Manual of Mental Disorders, CBCL Child Behavior CheckList, CPT Conners’ Continuous Performance Test, SNAP Swanson, Nolan, and Pelham, ODD Oppositional Defiant Disorder, BSID Bayley Scales of Infant Development, BASC Behavior Assessment System for Children, WISC Wechsler Intelligence Scale for Children, WPPSI-R Wechsler Preschool and Primary Scale of Intelligence-Revised, SDQ Strengths and Difficulties Questionnaire, IBQ Infant Behavior Questionnaire, TBAQ Toddler Behavior Assessment Questionnaire, VMWM Virtual Morris Water Maze, SMS Social Maturity Scale, IDS Intelligence and Development Scales, NEPSY Neuropsychological Assessment, WCST Wisconsin Card Sort Task-64, ENI Evaluación neuropsicológica del niño, SRS Social Responsiveness Scale, SRP Self-Report ofPersonality, CADS Conners’-ADHD/DSM-IV Scales, ASQ Ages and Stages Questionnaire, CPRS Conners’ Parent Rating Scale-Revised: Long Form, LMW Low Molecular Weight, and HMW High Molecular Weight.

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
