# Peer review of "A Review of Recent Studies on Bisphenol A and Phthalate Exposures and Child Neurodevelopment"

_ijerph, 2021, doi:10.3390/ijerph18073585_

Round 1

Reviewer 1 Report

This is a great and timely review. Authors have made excellent effort to put together the literature in context. 

The adverse effects of bisphenol A and phthalates on prenatal and postnatal conditions with data support is well articulated. But there are few parts in the manuscript that lacks clear explanation, instead read generic. I have marked those places in the reviewed file. Authors should thoughtfully take it into consideration and revise the article. 

Author Response

Thank you for the comments. Please take a look at below point by point response along with revised manuscript.

  1. Rewrite this section critically. Brief review of studies prior to the past 5 years

The section was rewritten. More details of the previous review articles were added. In addition, the limitations of previous articles and the importance of publishing review article of the recent years were described.

  1. Add the date source here. “The number of articles including the terms “bisphenol A” or “phthalate”, ”child” and “neurodevelopment” prior to the past 5 years was only 11, and 18, respectively.

The date and the data source was added in the parentheses. (January, 2021, PubMed)

  1. Rephrase the sentence slightly. “Thus, reviewing focused on recent findings may be worthy of note.”

The sentence was rephrased as “Thus, reviewing focused on the recent findings may be more helpful to understand what is known about the association between exposures and child neurodevelopmental outcomes and what are the issues and future directions.”

  1. The informations are more generic than anything specifically informing. What type of association between prenatal and childhood exposure to bisphenol A and pathalates is being discussed here? Instead of saying adverse neurobehavioural outcomes, it would be more explanatory to describe those particular outcomes.

Please see the response to the bullet No1. The whole section was rewritten trying to provide more specific information and details of previous studies rather than being general.

  1. So far these vague mention like poor or adverse outcomes seems unclear. “Poor developmental outcomes”

The part was described more specific as “poor developmental outcomes such as decreased IQ, poorer memory, ASD, ADHD and other behavioral problems” and “poorer outcomes including developmental delay, and disorders”.

  1. Non availability of several articles in the pubmed constraints the method here. Authors should consider referring to other major databases including Google Scholar, Medline, and so.

Other major databases were also mentioned.

  1. Maybe include the years in parentheses? “5 years”

(2015-2020) was added.

Reviewer 2 Report

Very comprehensive review on a significant, clinically-related subject.

A few comments to be taken into consideration:

  1. Articles published in 2020 should be included.
  2. The abstract is somehow misleading in stating " Overall, this review suggests that prenatal...." while in the manuscript it is clear that postnatal exposuse has been linked to adverse outcomes. Please rephrase this part of the abstract.

Additional comments:

Based on the information included in the manuscript, could certain phthalates be linked to specific behavioral adverse outcomes?

In section 2.7.1, the sentence "Gestational bisphenol A...are sex-specific" is out of its correct place, in section 2.7.2

Author Response

Very comprehensive review on a significant, clinically-related subject.

A few comments to be taken into consideration:

Thank you for the kind comments. Please see below response for your comments.

  1. Articles published in 2020 should be included.

Regarding BPA, 4 longitudinal and 1 case-control studies, regarding phthalates, 4 longitudinal studies published in 2020 were added. Please see the tables for the details of the studies. This was also mentioned in method section as “The search was conducted initially in December 2019 and an additional search in December 2020 using the PubMed, Google Scholar, and Medline database. The terms included “BPA” or “bisphenol A” and “neurodevelopment” or “neurodevelopmental disorder.” Similarly, terms included "phthalate" and “neurodevelopment” or “neurodevelopmental disorder.” Of these, 24 articles for bisphenol A and 30 articles for phthalate in association with neurodevelopmental outcomes were included.”

  1. The abstract is somehow misleading in stating " Overall, this review suggests that prenatal...." while in the manuscript it is clear that postnatal exposure has been linked to adverse outcomes. Please rephrase this part of the abstract.

The authors did not agree to say that postnatal exposure has been linked to adverse outcomes since number of studies examined postnatal exposure effects. However, from this review bisphenol A exposure and ADHD symptoms among boys seemed to be consistent finding, therefore the following was added to the abstract. “The evidence is still limited however, ADHD symptoms, especially among boys, constantly suggested association with both prenatal and concurrent exposure to bisphenol A.”

  1. Based on the information included in the manuscript, could certain phthalates be linked to specific behavioral adverse outcomes?

As discussed in the manuscript, there is no clear pattern of association between phthalate exposures and adverse outcomes in child neurodevelopment.   

  1. In section 2.7.1, the sentence "Gestational bisphenol A...are sex-specific" is out of its correct place, in section 2.7.2

The sentence was moved to the right place.

Round 2

Reviewer 1 Report

Authors have addressed all comments raised by reviewers. The manuscript is ready for acceptance.